# GLUE: A Multi-Task Benchmark and Analysis Platform for Natural Language Understanding

**Alex Wang**[1], **Amanpreet Singh**[1], **Julian Michael**[2], **Felix Hill**[3],
**Omer Levy**[2] **& Samuel R. Bowman**[1]
[1]Courant Institute of Mathematical Sciences, New York University
[2]Paul G. Allen School of Computer Science & Engineering, University of Washington
[3]DeepMind
{alexwang,amanpreet,bowman}@nyu.edu
{julianjm,omerlevy}@cs.washington.edu
felixhill@google.com

## Abstract

For natural language understanding (NLU) technology to be maximally useful, it must be able to process language in a way that is not exclusive to a single task, genre, or dataset. In pursuit of this objective, we introduce the General Language Understanding Evaluation (GLUE) benchmark, a collection of tools for evaluating the performance of models across a diverse set of existing NLU tasks. By including tasks with limited training data, GLUE is designed to favor and encourage models that share general linguistic knowledge across tasks. GLUE also includes a hand-crafted diagnostic test suite that enables detailed linguistic analysis of models. We evaluate baselines based on current methods for transfer and representation learning and find that multi-task training on all tasks performs better than training a separate model per task. However, the low absolute performance of our best model indicates the need for improved general NLU systems.

## 1 Introduction

The human ability to understand language is *general*, *flexible*, and *robust*. In contrast, most NLU models above the word level are designed for a specific task and struggle with out-of-domain data. If we aspire to develop models with understanding beyond the detection of superficial correspondences between inputs and outputs, then it is critical to develop a more unified model that can learn to execute a range of different linguistic tasks in different domains.

To facilitate research in this direction, we present the General Language Understanding Evaluation (GLUE) benchmark: a collection of NLU tasks including question answering, sentiment analysis, and textual entailment, and an associated online platform for model evaluation, comparison, and analysis. GLUE does not place any constraints on model architecture beyond the ability to process single-sentence and sentence-pair inputs and to make corresponding predictions. For some GLUE tasks, training data is plentiful, but for others it is limited or fails to match the genre of the test set. GLUE therefore favors models that can learn to represent linguistic knowledge in a way that facilitates sample-efficient learning and effective knowledge-transfer across tasks. None of the datasets in GLUE were created from scratch for the benchmark; we rely on preexisting datasets because they have been implicitly agreed upon by the NLP community as challenging and interesting. Four of the datasets feature privately-held test data, which will be used to ensure that the benchmark is used fairly.[1]

To understand the types of knowledge learned by models and to encourage linguistic-meaningful solution strategies, GLUE also includes a set of hand-crafted analysis examples for probing trained models. This dataset is designed to highlight common challenges, such as the use of world knowledge and logical operators, that we expect models must handle to robustly solve the tasks.

---

[1]To evaluate on the private test data, users of the benchmark must submit to `gluebenchmark.com`

| Corpus | \|Train\| | \|Test\| | Task | Metrics | Domain |
|--------|-----------|----------|------|---------|--------|
| | | | Single-Sentence Tasks | | |
| CoLA | 8.5k | **1k** | acceptability | Matthews corr. | misc. |
| SST-2 | 67k | 1.8k | sentiment | acc. | movie reviews |
| | | | Similarity and Paraphrase Tasks | | |
| MRPC | 3.7k | 1.7k | paraphrase | acc./F1 | news |
| STS-B | 7k | 1.4k | sentence similarity | Pearson/Spearman corr. | misc. |
| QQP | 364k | **391k** | paraphrase | acc./F1 | social QA questions |
| | | | Inference Tasks | | |
| MNLI | 393k | **20k** | NLI | matched acc./mismatched acc. | misc. |
| QNLI | 105k | 5.4k | QA/NLI | acc. | Wikipedia |
| RTE | 2.5k | 3k | NLI | acc. | news, Wikipedia |
| WNLI | 634 | **146** | coreference/NLI | acc. | fiction books |

Table 1: Task descriptions and statistics. All tasks are single sentence or sentence pair classification, except STS-B, which is a regression task. MNLI has three classes; all other classification tasks have two. Test sets shown in bold use labels that have never been made public in any form.

To better understand the challenged posed by GLUE, we conduct experiments with simple baselines and state-of-the-art sentence representation models. We find that unified multi-task trained models slightly outperform comparable models trained on each task separately. Our best multi-task model makes use of ELMo (Peters et al., 2018), a recently proposed pre-training technique. However, this model still achieves a fairly low absolute score. Analysis with our diagnostic dataset reveals that our baseline models deal well with strong lexical signals but struggle with deeper logical structure.

In summary, we offer: (i) A suite of nine sentence or sentence-pair NLU tasks, built on established annotated datasets and selected to cover a diverse range of text genres, dataset sizes, and degrees of difficulty. (ii) An online evaluation platform and leaderboard, based primarily on privately-held test data. The platform is model-agnostic, and can evaluate any method capable of producing results on all nine tasks. (iii) An expert-constructed diagnostic evaluation dataset. (iv) Baseline results for several major existing approaches to sentence representation learning.

## 2 RELATED WORK

Collobert et al. (2011) used a multi-task model with a shared sentence understanding component to jointly learn POS tagging, chunking, named entity recognition, and semantic role labeling. More recent work has explored using labels from core NLP tasks to supervise training of lower levels of deep neural networks (Søgaard & Goldberg, 2016; Hashimoto et al., 2017) and automatically learning cross-task sharing mechanisms for multi-task learning (Ruder et al., 2017).

Beyond multi-task learning, much work in developing general NLU systems has focused on sentence-to-vector encoders (Le & Mikolov, 2014; Kiros et al., 2015, i.a.), leveraging unlabeled data (Hill et al., 2016; Peters et al., 2018), labeled data (Conneau & Kiela, 2018; McCann et al., 2017), and combinations of these (Collobert et al., 2011; Subramanian et al., 2018). In this line of work, a standard evaluation practice has emerged, recently codified as SentEval (Conneau et al., 2017; Conneau & Kiela, 2018). Like GLUE, SentEval relies on a set of existing classification tasks involving either one or two sentences as inputs. Unlike GLUE, SentEval only evaluates sentence-to-vector encoders, making it well-suited for evaluating models on tasks involving sentences *in isolation*. However, cross-sentence contextualization and alignment are instrumental in achieving state-of-the-art performance on tasks such as machine translation (Bahdanau et al., 2015; Vaswani et al., 2017), question answering (Seo et al., 2017), and natural language inference (Rocktäschel et al., 2016). GLUE is designed to facilitate the development of these methods: It is model-agnostic, allowing for any kind of representation or contextualization, including models that use no explicit vector or symbolic representations for sentences whatsoever.

GLUE also diverges from SentEval in the selection of evaluation tasks that are included in the suite. Many of the SentEval tasks are closely related to sentiment analysis, such as MR (Pang & Lee,

2005), SST (Socher et al., 2013), CR (Hu & Liu, 2004), and SUBJ (Pang & Lee, 2004). Other tasks are so close to being solved that evaluation on them is relatively uninformative, such as MPQA (Wiebe et al., 2005) and TREC question classification (Voorhees et al., 1999). In GLUE, we attempt to construct a benchmark that is both diverse and difficult.

McCann et al. (2018) introduce decaNLP, which also scores NLP systems based on their performance on multiple datasets. Their benchmark recasts the ten evaluation tasks as question answering, converting tasks like summarization and text-to-SQL semantic parsing into question answering using automatic transformations. That benchmark lacks the leaderboard and error analysis toolkit of GLUE, but more importantly, we see it as pursuing a more ambitious but less immediately practical goal: While GLUE rewards methods that yield good performance on a circumscribed set of tasks using methods like those that are currently used for those tasks, their benchmark rewards systems that make progress toward their goal of unifying all of NLU under the rubric of question answering.

## 3 TASKS

GLUE is centered on nine English sentence understanding tasks, which cover a broad range of domains, data quantities, and difficulties. As the goal of GLUE is to spur development of generalizable NLU systems, we design the benchmark such that good performance should require a model to share substantial knowledge (e.g., trained parameters) across all tasks, while still maintaining some task-specific components. Though it is possible to train a single model for each task with no pretraining or other outside sources of knowledge and evaluate the resulting set of models on this benchmark, we expect that our inclusion of several data-scarce tasks will ultimately render this approach uncompetitive. We describe the tasks below and in Table 1. Appendix A includes additional details. Unless otherwise mentioned, tasks are evaluated on accuracy and are balanced across classes.

### 3.1 SINGLE-SENTENCE TASKS

**CoLA** The Corpus of Linguistic Acceptability (Warstadt et al., 2018) consists of English acceptability judgments drawn from books and journal articles on linguistic theory. Each example is a sequence of words annotated with whether it is a grammatical English sentence. Following the authors, we use Matthews correlation coefficient (Matthews, 1975) as the evaluation metric, which evaluates performance on unbalanced binary classification and ranges from -1 to 1, with 0 being the performance of uninformed guessing. We use the standard test set, for which we obtained private labels from the authors. We report a single performance number on the combination of the in- and out-of-domain sections of the test set.

**SST-2** The Stanford Sentiment Treebank (Socher et al., 2013) consists of sentences from movie reviews and human annotations of their sentiment. The task is to predict the sentiment of a given sentence. We use the two-way (*positive/negative*) class split, and use only sentence-level labels.

### 3.2 SIMILARITY AND PARAPHRASE TASKS

**MRPC** The Microsoft Research Paraphrase Corpus (Dolan & Brockett, 2005) is a corpus of sentence pairs automatically extracted from online news sources, with human annotations for whether the sentences in the pair are semantically equivalent. Because the classes are imbalanced (68% positive), we follow common practice and report both accuracy and F1 score.

**QQP** The Quora Question Pairs[2] dataset is a collection of question pairs from the community question-answering website Quora. The task is to determine whether a pair of questions are semantically equivalent. As in MRPC, the class distribution in QQP is unbalanced (63% negative), so we report both accuracy and F1 score. We use the standard test set, for which we obtained private labels from the authors. We observe that the test set has a different label distribution than the training set.

**STS-B** The Semantic Textual Similarity Benchmark (Cer et al., 2017) is a collection of sentence pairs drawn from news headlines, video and image captions, and natural language inference data.

---

[2] data.quora.com/First-Quora-Dataset-Release-Question-Pairs

| Coarse-Grained Categories | Fine-Grained Categories |
|---|---|
| Lexical Semantics | Lexical Entailment, Morphological Negation, Factivity, Symmetry/Collectivity, Redundancy, Named Entities, Quantifiers |
| Predicate-Argument Structure | Core Arguments, Prepositional Phrases, Ellipsis/Implicits, Anaphora/Coreference Active/Passive, Nominalization, Genitives/Partitives, Datives, Relative Clauses, Coordination Scope, Intersectivity, Restrictivity |
| Logic | Negation, Double Negation, Intervals/Numbers, Conjunction, Disjunction, Conditionals, Universal, Existential, Temporal, Upward Monotone, Downward Monotone, Non-Monotone |
| Knowledge | Common Sense, World Knowledge |

Table 2: The types of linguistic phenomena annotated in the diagnostic dataset, organized under four major categories. For a description of each phenomenon, see Appendix E.

Each pair is human-annotated with a similarity score from 1 to 5; the task is to predict these scores. Follow common practice, we evaluate using Pearson and Spearman correlation coefficients.

## 3.3 INFERENCE TASKS

**MNLI** The Multi-Genre Natural Language Inference Corpus (Williams et al., 2018) is a crowd-sourced collection of sentence pairs with textual entailment annotations. Given a premise sentence and a hypothesis sentence, the task is to predict whether the premise entails the hypothesis (*entailment*), contradicts the hypothesis (*contradiction*), or neither (*neutral*). The premise sentences are gathered from ten different sources, including transcribed speech, fiction, and government reports. We use the standard test set, for which we obtained private labels from the authors, and evaluate on both the *matched* (in-domain) and *mismatched* (cross-domain) sections. We also use and recommend the SNLI corpus (Bowman et al., 2015) as 550k examples of auxiliary training data.

**QNLI** The Stanford Question Answering Dataset (Rajpurkar et al. 2016) is a question-answering dataset consisting of question-paragraph pairs, where one of the sentences in the paragraph (drawn from Wikipedia) contains the answer to the corresponding question (written by an annotator). We convert the task into sentence pair classification by forming a pair between each question and each sentence in the corresponding context, and filtering out pairs with low lexical overlap between the question and the context sentence. The task is to determine whether the context sentence contains the answer to the question. This modified version of the original task removes the requirement that the model select the exact answer, but also removes the simplifying assumptions that the answer is always present in the input and that lexical overlap is a reliable cue. This process of recasting existing datasets into NLI is similar to methods introduced in White et al. (2017) and expanded upon in Demszky et al. (2018). We call the converted dataset QNLI (Question-answering NLI).[3]

**RTE** The Recognizing Textual Entailment (RTE) datasets come from a series of annual textual entailment challenges. We combine the data from RTE1 (Dagan et al., 2006), RTE2 (Bar Haim et al., 2006), RTE3 (Giampiccolo et al., 2007), and RTE5 (Bentivogli et al., 2009).[4] Examples are constructed based on news and Wikipedia text. We convert all datasets to a two-class split, where for three-class datasets we collapse *neutral* and *contradiction* into *not_entailment*, for consistency.

**WNLI** The Winograd Schema Challenge (Levesque et al., 2011) is a reading comprehension task in which a system must read a sentence with a pronoun and select the referent of that pronoun from a list of choices. The examples are manually constructed to foil simple statistical methods: Each one is contingent on contextual information provided by a single word or phrase in the sentence. To convert the problem into sentence pair classification, we construct sentence pairs by replacing the ambiguous pronoun with each possible referent. The task is to predict if the sentence with the

---

[3]An earlier release of QNLI had an artifact where the task could be modeled and solved as an easier task than we describe here. We have since released an updated version of QNLI that removes this possibility.

[4]RTE4 is not publicly available, while RTE6 and RTE7 do not fit the standard NLI task.

| Tags | Sentence 1 | Sentence 2 | Fwd | Bwd |
|---|---|---|---|---|
| *Lexical Entailment (Lexical Semantics), Downward Monotone (Logic)* | The timing of the meeting has not been set, according to a Starbucks spokesperson. | The timing of the meeting has not been considered, according to a Starbucks spokesperson. | N | E |
| *Universal Quantifiers (Logic)* | Our deepest sympathies are with all those affected by this accident. | Our deepest sympathies are with a victim who was affected by this accident. | E | N |
| *Quantifiers (Lexical Semantics), Double Negation (Logic)* | I have never seen a hummingbird not flying. | I have never seen a hummingbird. | N | E |

Table 3: Examples from the diagnostic set. *Fwd* (resp. *Bwd*) denotes the label when sentence 1 (resp. sentence 2) is the premise. Labels are *entailment* (E), *neutral* (N), or *contradiction* (C). Examples are tagged with the phenomena they demonstrate, and each phenomenon belongs to one of four broad categories (in parentheses).

pronoun substituted is entailed by the original sentence. We use a small evaluation set consisting of new examples derived from fiction books[5] that was shared privately by the authors of the original corpus. While the included training set is balanced between two classes, the test set is imbalanced between them (65% not entailment). Also, due to a data quirk, the development set is *adversarial*: hypotheses are sometimes shared between training and development examples, so if a model memorizes the training examples, they will predict the wrong label on corresponding development set example. As with QNLI, each example is evaluated separately, so there is not a systematic correspondence between a model's score on this task and its score on the unconverted original task. We call converted dataset WNLI (Winograd NLI).

### 3.4 EVALUATION

The GLUE benchmark follows the same evaluation model as SemEval and Kaggle. To evaluate a system on the benchmark, one must run the system on the provided test data for the tasks, then upload the results to the website `gluebenchmark.com` for scoring. The benchmark site shows per-task scores and a macro-average of those scores to determine a system's position on the leaderboard. For tasks with multiple metrics (e.g., accuracy and F1), we use an unweighted average of the metrics as the score for the task when computing the overall macro-average. The website also provides fine- and coarse-grained results on the diagnostic dataset. See Appendix D for details.

## 4 DIAGNOSTIC DATASET

Drawing inspiration from the FraCaS suite (Cooper et al., 1996) and the recent Build-It-Break-It competition (Ettinger et al., 2017), we include a small, manually-curated test set for the analysis of system performance. While the main benchmark mostly reflects an application-driven distribution of examples, our diagnostic dataset highlights a pre-defined set of phenomena that we believe are interesting and important for models to capture. We show the full set of phenomena in Table 2.

Each diagnostic example is an NLI sentence pair with tags for the phenomena demonstrated. The NLI task is well-suited to this kind of analysis, as it can easily evaluate the full set of skills involved in (ungrounded) sentence understanding, from resolution of syntactic ambiguity to pragmatic reasoning with world knowledge. We ensure the data is reasonably diverse by producing examples for a variety of linguistic phenomena and basing our examples on naturally-occurring sentences from several domains (news, Reddit, Wikipedia, academic papers). This approaches differs from that of FraCaS, which was designed to test linguistic theories with a minimal and uniform set of examples. A sample from our dataset is shown in Table 3.

---

[5]See similar examples at `cs.nyu.edu/faculty/davise/papers/WinogradSchemas/WS.html`

**Annotation Process** We begin with a target set of phenomena, based roughly on those used in the FraCaS suite (Cooper et al., 1996). We construct each example by locating a sentence that can be easily made to demonstrate a target phenomenon, and editing it in two ways to produce an appropriate sentence pair. We make minimal modifications so as to maintain high lexical and structural overlap within each sentence pair and limit superficial cues. We then label the inference relationships between the sentences, considering each sentence alternatively as the premise, producing two labeled examples for each pair (1100 total). Where possible, we produce several pairs with different labels for a single source sentence, to have minimal sets of sentence pairs that are lexically and structurally very similar but correspond to different entailment relationships. The resulting labels are 42% *entailment*, 35% *neutral*, and 23% *contradiction*.

**Evaluation** Since the class distribution in the diagnostic set is not balanced, we use $R_3$ (Gorodkin, 2004), a three-class generalization of the Matthews correlation coefficient, for evaluation.

In light of recent work showing that crowdsourced data often contains artifacts which can be exploited to perform well without solving the intended task (Schwartz et al., 2017; Poliak et al., 2018; Tsuchiya, 2018, i.a.), we audit the data for such artifacts. We reproduce the methodology of Gururangan et al. (2018), training two fastText classifiers (Joulin et al., 2016) to predict entailment labels on SNLI and MNLI using only the hypothesis as input. The models respectively get near-chance accuracies of 32.7% and 36.4% on our diagnostic data, showing that the data does not suffer from such artifacts.

To establish human baseline performance on the diagnostic set, we have six NLP researchers annotate 50 sentence pairs (100 entailment examples) randomly sampled from the diagnostic set. Inter-annotator agreement is high, with a Fleiss's $\kappa$ of 0.73. The average $R_3$ score among the annotators is 0.80, much higher than any of the baseline systems described in Section 5.

**Intended Use** The diagnostic examples are hand-picked to address certain phenomena, and NLI is a task with no natural input distribution, so we do not expect performance on the diagnostic set to reflect overall performance or generalization in downstream applications. Performance on the analysis set should be compared between models but not between categories. The set is provided not as a benchmark, but as an analysis tool for error analysis, qualitative model comparison, and development of adversarial examples.

## 5 BASELINES

For baselines, we evaluate a multi-task learning model trained on the GLUE tasks, as well as several variants based on recent pre-training methods. We briefly describe them here. See Appendix B for details. We implement our models in the AllenNLP library (Gardner et al., 2017). Original code for the baselines is available at `https://github.com/nyu-mll/GLUE-baselines` and a newer version is available at `https://github.com/jsalt18-sentence-repl/jiant`.

**Architecture** Our simplest baseline architecture is based on sentence-to-vector encoders, and sets aside GLUE's ability to evaluate models with more complex structures. Taking inspiration from Conneau et al. (2017), the model uses a two-layer, 1500D (per direction) BiLSTM with max pooling and 300D GloVe word embeddings (840B Common Crawl version; Pennington et al., 2014). For single-sentence tasks, we encode the sentence and pass the resulting vector to a classifier. For sentence-pair tasks, we encode sentences independently to produce vectors $u, v$, and pass $[u; v; |u - v|; u * v]$ to a classifier. The classifier is an MLP with a 512D hidden layer.

We also consider a variant of our model which for sentence pair tasks uses an attention mechanism inspired by Seo et al. (2017) between all pairs of words, followed by a second BiLSTM with max pooling. By explicitly modeling the interaction between sentences, these models fall outside the sentence-to-vector paradigm.

**Pre-Training** We augment our base model with two recent methods for pre-training: ELMo and CoVe. We use existing trained models for both.

ELMo uses a pair of two-layer neural language models trained on the Billion Word Benchmark (Chelba et al., 2013). Each word is represented by a contextual embedding, produced by taking a

| Model | Avg | Single Sentence | | Similarity and Paraphrase | | | Natural Language Inference | | | |
| | | CoLA | SST-2 | MRPC | QQP | STS-B | MNLI | QNLI | RTE | WNLI |
|---|---|---|---|---|---|---|---|---|---|---|
| | | | | Single-Task Training | | | | | | |
| BiLSTM | 63.9 | 15.7 | 85.9 | 69.3/79.4 | 81.7/61.4 | 66.0/62.8 | 70.3/70.8 | 75.7 | 52.8 | **65.1** |
| +ELMo | 66.4 | **35.0** | 90.2 | 69.0/80.8 | 85.7/65.6 | 64.0/60.2 | 72.9/73.4 | 71.7 | 50.1 | **65.1** |
| +CoVe | 64.0 | 14.5 | 88.5 | 73.4/81.4 | 83.3/59.4 | 67.2/64.1 | 64.5/64.8 | 75.4 | 53.5 | **65.1** |
| +Attn | 63.9 | 15.7 | 85.9 | 68.5/80.3 | 83.5/62.9 | 59.3/55.8 | 74.2/73.8 | 77.2 | 51.9 | **65.1** |
| +Attn, ELMo | 66.5 | **35.0** | 90.2 | 68.8/80.2 | **86.5/66.1** | 55.5/52.5 | 76.9/76.7 | 76.7 | 50.4 | **65.1** |
| +Attn, CoVe | 63.2 | 14.5 | 88.5 | 68.6/79.7 | 84.1/60.1 | 57.2/53.6 | 71.6/71.5 | 74.5 | 52.7 | **65.1** |
| | | | | Multi-Task Training | | | | | | |
| BiLSTM | 64.2 | 11.6 | 82.8 | 74.3/81.8 | 84.2/62.5 | 70.3/67.8 | 65.4/66.1 | 74.6 | 57.4 | **65.1** |
| +ELMo | 67.7 | 32.1 | 89.3 | **78.0/84.7** | 82.6/61.1 | 67.2/67.9 | 70.3/67.8 | 75.5 | 57.4 | **65.1** |
| +CoVe | 62.9 | 18.5 | 81.9 | 71.5/78.7 | 84.9/60.6 | 64.4/62.7 | 65.4/65.7 | 70.8 | 52.7 | **65.1** |
| +Attn | 65.6 | 18.6 | 83.0 | 76.2/83.9 | 82.4/60.1 | 72.8/70.5 | 67.6/68.3 | 74.3 | 58.4 | **65.1** |
| +Attn, ELMo | **70.0** | 33.6 | 90.4 | **78.0**/84.4 | 84.3/63.1 | 74.2/72.3 | 74.1/74.5 | **79.8** | 58.9 | **65.1** |
| +Attn, CoVe | 63.1 | 8.3 | 80.7 | 71.8/80.0 | 83.4/60.5 | 69.8/68.4 | 68.1/68.6 | 72.9 | 56.0 | **65.1** |
| | | | | Pre-Trained Sentence Representation Models | | | | | | |
| CBoW | 58.9 | 0.0 | 80.0 | 73.4/81.5 | 79.1/51.4 | 61.2/58.7 | 56.0/56.4 | 72.1 | 54.1 | **65.1** |
| Skip-Thought | 61.3 | 0.0 | 81.8 | 71.7/80.8 | 82.2/56.4 | 71.8/69.7 | 62.9/62.8 | 72.9 | 53.1 | **65.1** |
| InferSent | 63.9 | 4.5 | 85.1 | 74.1/81.2 | 81.7/59.1 | 75.9/75.3 | 66.1/65.7 | 72.7 | 58.0 | **65.1** |
| DisSent | 62.0 | 4.9 | 83.7 | 74.1/81.7 | 82.6/59.5 | 66.1/64.8 | 58.7/59.1 | 73.9 | 56.4 | **65.1** |
| GenSen | 66.2 | 7.7 | 83.1 | 76.6/83.0 | 82.9/59.8 | 79.3/79.2 | 71.4/71.3 | 78.6 | **59.2** | **65.1** |

Table 4: Baseline performance on the GLUE task test sets. For MNLI, we report accuracy on the matched and mismatched test sets. For MRPC and Quora, we report accuracy and F1. For STS-B, we report Pearson and Spearman correlation. For CoLA, we report Matthews correlation. For all other tasks we report accuracy. All values are scaled by 100. A similar table is presented on the online platform.

linear combination of the corresponding hidden states of each layer of the two models. We follow the authors' recommendations[6] and use ELMo embeddings in place of any other embeddings.

CoVe (McCann et al., 2017) uses a two-layer BiLSTM encoder originally trained for English-to-German translation. The CoVe vector of a word is the corresponding hidden state of the top-layer LSTM. As in the original work, we concatenate the CoVe vectors to the GloVe word embeddings.

**Training** We train our models with the BiLSTM sentence encoder and post-attention BiLSTMs shared across tasks, and classifiers trained separately for each task. For each training update, we sample a task to train with a probability proportional to the number of training examples for each task. We train our models with Adam (Kingma & Ba, 2015) with initial learning rate $10^{-4}$ and batch size 128. We use the macro-average score as the validation metric and stop training when the learning rate drops below $10^{-5}$ or performance does not improve after 5 validation checks.

We also train a set of single-task models, which are configured and trained identically, but share no parameters. To allow for fair comparisons with the multi-task analogs, we do not tune parameter or training settings for each task, so these single-task models do not generally represent the state of the art for each task.

**Sentence Representation Models** Finally, we evaluate the following trained sentence-to-vector encoder models using our benchmark: average bag-of-words using GloVe embeddings (CBoW), Skip-Thought (Kiros et al., 2015), InferSent (Conneau et al., 2017), DisSent (Nie et al., 2017), and GenSen (Subramanian et al., 2018). For these models, we only train task-specific classifiers on the representations they produce.

---

[6]github.com/allenai/allennlp/blob/master/tutorials/how_to/elmo.md

| Model | All | Coarse-Grained | | | | Fine-Grained | | | | | |
|---|---|---|---|---|---|---|---|---|---|---|---|
| | | LS | PAS | L | K | UQuant | MNeg | 2Neg | Coref | Restr | Down |
| *Single-Task Training* | | | | | | | | | | | |
| BiLSTM | 21 | 25 | 24 | 16 | 16 | 70 | 53 | 4 | 21 | -15 | **12** |
| +ELMo | 20 | 20 | 21 | 14 | 17 | 70 | 20 | **42** | 33 | -26 | -3 |
| +CoVe | 21 | 19 | 23 | 20 | 18 | 71 | 47 | -1 | 33 | -15 | 8 |
| +Attn | 25 | 24 | 30 | 20 | 14 | 50 | 47 | 21 | 38 | -8 | -3 |
| +Attn, ELMo | **28** | **30** | **35** | **23** | 14 | **85** | 20 | 42 | 33 | -26 | -3 |
| +Attn, CoVe | 24 | 29 | 29 | 18 | 12 | 77 | 50 | 1 | 18 | -1 | **12** |
| *Multi-Task Training* | | | | | | | | | | | |
| BiLSTM | 20 | 13 | 24 | 14 | 22 | 71 | 17 | -8 | 31 | -15 | 8 |
| +ELMo | 21 | 20 | 21 | 19 | 21 | 71 | **60** | 2 | 22 | 0 | **12** |
| +CoVe | 18 | 15 | 11 | 18 | **27** | 71 | 40 | 7 | **40** | 0 | 8 |
| +Attn | 18 | 13 | 24 | 11 | 16 | 71 | 1 | -12 | 31 | -15 | 8 |
| +Attn, ELMo | 22 | 18 | 26 | 13 | 19 | 70 | 27 | 5 | 31 | -26 | -3 |
| +Attn, CoVe | 18 | 16 | 25 | 16 | 13 | 71 | 26 | -8 | 33 | **9** | 8 |
| *Pre-Trained Sentence Representation Models* | | | | | | | | | | | |
| CBoW | 9 | 6 | 13 | 5 | 10 | 3 | 0 | 13 | 28 | -15 | -11 |
| Skip-Thought | 12 | 2 | 23 | 11 | 9 | 61 | 6 | -2 | 30 | -15 | 0 |
| InferSent | 18 | 20 | 20 | 15 | 14 | 77 | 50 | -20 | 15 | -15 | -9 |
| DisSent | 16 | 16 | 19 | 13 | 15 | 70 | 43 | -11 | 20 | -36 | -09 |
| GenSen | 20 | 28 | 26 | 14 | 12 | 78 | 57 | 2 | 21 | -15 | **12** |

Table 5: Results on the diagnostic set. We report $R_3$ coefficients between gold and predicted labels, scaled by 100. The coarse-grained categories are *Lexical Semantics* (**LS**), *Predicate-Argument Structure* (**PAS**), *Logic* (**L**), and *Knowledge and Common Sense* (**K**). Our example fine-grained categories are *Universal Quantification* (**UQuant**), *Morphological Negation* (**MNeg**), *Double Negation* (**2Neg**), *Anaphora/Coreference* (**Coref**), *Restrictivity* (**Restr**), and *Downward Monotone* (**Down**).

## 6 BENCHMARK RESULTS

We train three runs of each model and evaluate the run with the best macro-average development set performance (see Table 6 in Appendix C). For single-task and sentence representation models, we evaluate the best run for each individual task. We present performance on the main benchmark tasks in Table 4.

We find that multi-task training yields better overall scores over single-task training amongst models using attention or ELMo. Attention generally has negligible or negative aggregate effect in single task training, but helps in multi-task training. We see a consistent improvement in using ELMo embeddings in place of GloVe or CoVe embeddings, particularly for single-sentence tasks. Using CoVe has mixed effects over using only GloVe.

Among the pre-trained sentence representation models, we observe fairly consistent gains moving from CBoW to Skip-Thought to Infersent and GenSen. Relative to the models trained directly on the GLUE tasks, InferSent is competitive and GenSen outperforms all but the two best.

Looking at results per task, we find that the sentence representation models substantially underperform on CoLA compared to the models directly trained on the task. On the other hand, for STS-B, models trained directly on the task lag significantly behind the performance of the best sentence representation model. Finally, there are tasks for which no model does particularly well. On WNLI, no model exceeds most-frequent-class guessing (65.1%) and we substitute the model predictions for the most-frequent baseline. On RTE and in aggregate, even our best baselines leave room for improvement. These early results indicate that solving GLUE is beyond the capabilities of current models and methods.

## 7 ANALYSIS

We analyze the baselines by evaluating each model's MNLI classifier on the diagnostic set to get a better sense of their linguistic capabilities. Results are presented in Table 5.

**Coarse Categories**    Overall performance is low for all models: The highest total score of 28 still denotes poor absolute performance. Performance tends to be higher on Predicate-Argument Structure and lower on Logic, though numbers are not closely comparable across categories. Unlike on the main benchmark, the multi-task models are almost always outperformed by their single-task counterparts. This is perhaps unsurprising, since with our simple multi-task training regime, there is likely some destructive interference between MNLI and the other tasks. The models trained on the GLUE tasks largely outperform the pretrained sentence representation models, with the exception of GenSen. Using attention has a greater influence on diagnostic scores than using ELMo or CoVe, which we take to indicate that attention is especially important for generalization in NLI.

**Fine-Grained Subcategories**    Most models handle universal quantification relatively well. Looking at relevant examples, it seems that relying on lexical cues such as "all" often suffices for good performance. Similarly, lexical cues often provide good signal in morphological negation examples.

We observe varying weaknesses between models. Double negation is especially difficult for the GLUE-trained models that only use GloVe embeddings. This is ameliorated by ELMo, and to some degree CoVe. Also, attention has mixed effects on overall results, and models with attention tend to struggle with downward monotonicity. Examining their predictions, we found that the models are sensitive to hypernym/hyponym substitution and word deletion as a signal of entailment, but predict it in the wrong direction (as if the substituted/deleted word were in an upward monotone context). This is consistent with recent findings by McCoy & Linzen (2019) that these systems use the subsequence relation between premise and hypothesis as a heuristic shortcut. Restrictivity examples, which often depend on nuances of quantifier scope, are especially difficult for almost all models.

Overall, there is evidence that going beyond sentence-to-vector representations, e.g. with an attention mechanism, might aid performance on out-of-domain data, and that transfer methods like ELMo and CoVe encode linguistic information specific to their supervision signal. However, increased representational capacity may lead to overfitting, such as the failure of attention models in downward monotone contexts. We expect that our platform and diagnostic dataset will be useful for similar analyses in the future, so that model designers can better understand their models' generalization behavior and implicit knowledge.

## 8 CONCLUSION

We introduce GLUE, a platform and collection of resources for evaluating and analyzing natural language understanding systems. We find that, in aggregate, models trained jointly on our tasks see better performance than the combined performance of models trained for each task separately. We confirm the utility of attention mechanisms and transfer learning methods such as ELMo in NLU systems, which combine to outperform the best sentence representation models on the GLUE benchmark, but still leave room for improvement. When evaluating these models on our diagnostic dataset, we find that they fail (often spectacularly) on many linguistic phenomena, suggesting possible avenues for future work. In sum, the question of how to design general-purpose NLU models remains unanswered, and we believe that GLUE can provide fertile soil for addressing this challenge.

## ACKNOWLEDGMENTS

We thank Ellie Pavlick, Tal Linzen, Kyunghyun Cho, and Nikita Nangia for their comments on this work at its early stages, and we thank Ernie Davis, Alex Warstadt, and Quora's Nikhil Dandekar and Kornel Csernai for providing access to private evaluation data. This project has benefited from financial support to SB by Google, Tencent Holdings, and Samsung Research, and to AW from AdeptMind and an NSF Graduate Research Fellowship.

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

| Model | Avg | Single Sentence | | Similarity and Paraphrase | | | Natural Language Inference | | | |
| | | CoLA | SST-2 | MRPC | QQP | STS-B | MNLI | QNLI | RTE | WNLI |
|---|---|---|---|---|---|---|---|---|---|---|
| | | | | Single-Task Training | | | | | | |
| BiLSTM | 66.7 | 17.6 | 87.5 | 77.9/85.1 | 85.3/82.0 | 71.6/72.0 | 66.7 | 77.0 | 58.5 | 56.3 |
| +ELMo | 68.7 | 44.1 | 91.5 | 70.8/82.3 | 88.0/84.3 | 70.3/70.5 | 68.6 | 71.2 | 53.4 | 56.3 |
| +CoVe | 66.8 | 25.1 | 89.2 | 76.5/83.4 | 86.2/81.8 | 70.7/70.8 | 62.4 | 74.4 | 59.6 | 54.9 |
| +Attn | 66.9 | 17.6 | 87.5 | 72.8/82.9 | 87.7/83.9 | 66.6/66.7 | 70.0 | 77.2 | 58.5 | 60.6 |
| +Attn, ELMo | 67.9 | 44.1 | 91.5 | 71.1/82.1 | 87.8/83.6 | 57.9/56.1 | 72.4 | 75.2 | 52.7 | 56.3 |
| +Attn, CoVe | 65.6 | 25.1 | 89.2 | 72.8/82.4 | 86.1/81.3 | 59.4/58.0 | 67.9 | 72.5 | 58.1 | 57.7 |
| | | | | Multi-Task Training | | | | | | |
| BiLSTM | 60.0 | 18.6 | 82.3 | 75.0/82.7 | 84.4/79.3 | 69.0/66.9 | 65.6 | 74.9 | 59.9 | 9.9 |
| +ELMo | 63.1 | 26.4 | 90.9 | 80.2/86.7 | 84.2/79.7 | 72.9/71.5 | 67.4 | 76.0 | 55.6 | 14.1 |
| +CoVe | 59.3 | 9.8 | 82.0 | 73.8/81.0 | 83.4/76.6 | 64.5/61.9 | 65.5 | 70.4 | 52.7 | 32.4 |
| +Attn | 60.5 | 15.2 | 83.1 | 77.5/85.1 | 82.6/77.2 | 72.4/70.5 | 68.0 | 73.7 | 61.7 | 9.9 |
| +Attn, ELMo | 67.3 | 36.7 | 91.1 | 80.6/86.6 | 84.6/79.6 | 74.4/72.9 | 74.6 | 80.4 | 61.4 | 22.5 |
| +Attn, CoVe | 61.4 | 17.4 | 82.1 | 71.3/80.1 | 83.4/77.7 | 68.6/66.7 | 68.2 | 73.2 | 58.5 | 29.6 |
| | | | | Pre-Trained Sentence Representation Models | | | | | | |
| CBoW | 61.4 | 4.6 | 79.5 | 75.0/83.7 | 75.0/65.5 | 70.6/71.1 | 57.1 | 62.5 | 71.9 | 56.3 |
| Skip-Thought | 61.8 | 0.0 | 82.0 | 76.2/84.3 | 78.9/70.7 | 74.8/74.8 | 63.4 | 58.5 | 73.4 | 49.3 |
| InferSent | 65.7 | 8.6 | 83.9 | 76.5/84.1 | 81.7/75.9 | 80.2/80.4 | 67.8 | 63.5 | 71.5 | 56.3 |
| DisSent | 63.8 | 11.7 | 82.5 | 77.0/84.4 | 81.8/75.6 | 68.9/69.0 | 61.2 | 59.9 | 73.9 | 56.3 |
| GenSen | 67.8 | 10.3 | 87.2 | 80.4/86.2 | 82.6/76.6 | 81.3/81.8 | 71.4 | 62.5 | 78.4 | 56.3 |

Table 6: Baseline performance on the GLUE tasks' development sets. For MNLI, we report accuracy averaged over the matched and mismatched test sets. For MRPC and QQP, we report accuracy and F1. For STS-B, we report Pearson and Spearman correlation. For CoLA, we report Matthews correlation. For all other tasks we report accuracy. All values are scaled by 100.

Janyce Wiebe, Theresa Wilson, and Claire Cardie. Annotating expressions of opinions and emotions in language. In *Proceedings of the International Conference on Language Resources and Evaluation*, volume 39, pp. 165–210. Springer, 2005.

Adina Williams, Nikita Nangia, and Samuel R. Bowman. A broad-coverage challenge corpus for sentence understanding through inference. In *Proceedings of the North American Chapter of the Association for Computational Linguistics: Human Language Technologies*, 2018.

Yukun Zhu, Ryan Kiros, Rich Zemel, Ruslan Salakhutdinov, Raquel Urtasun, Antonio Torralba, and Sanja Fidler. Aligning books and movies: Towards story-like visual explanations by watching movies and reading books. In *Proceedings of the International Conference on Computer Vision*, pp. 19–27, 2015.

## A  ADDITIONAL BENCHMARK DETAILS

**QNLI**  To construct a balanced dataset, we select all pairs in which the most similar sentence to the question was *not* the answer sentence, as well as an equal amount of cases in which the correct sentence was the most similar to the question, but another distracting sentence was a close second. Our similarity metric is based on CBoW representations with pre-trained GloVe embeddings. This approach to converting pre-existing datasets into NLI format is closely related to recent work by White et al. (2017), as well as to the original motivation for textual entailment presented by Dagan et al. (2006). Both argue that many NLP tasks can be productively reduced to textual entailment.

## B   Additional Baseline Details

### B.1   Attention Mechanism

We implement our attention mechanism as follows: given two sequences of hidden states $u_1, u_2, \ldots, u_M$ and $v_1, v_2, \ldots, v_N$, we first compute matrix $H$ where $H_{ij} = u_i \cdot v_j$. For each $u_i$, we get attention weights $\alpha_i$ by taking a softmax over the $i^{th}$ row of $H$, and get the corresponding context vector $\tilde{v}_i = \sum_j \alpha_{ij} v_j$ by taking the attention-weighted sum of the $v_j$. We pass a second BiLSTM with max pooling over the sequence $[u_1; \tilde{v}_1], \ldots [u_M; \tilde{v}_M]$ to produce $u'$. We process the $v_j$ vectors analogously to obtain $v'$. Finally, we feed $[u'; v'; |u' - v'|; u' * v']$ into a classifier.

### B.2   Training

We train our models with the BiLSTM sentence encoder and post-attention BiLSTMs shared across tasks, and classifiers trained separately for each task. For each training update, we sample a task to train with a probability proportional to the number of training examples for each task. We scale each task's loss inversely proportional to the number of examples for that task, which we found to improve overall performance. We train our models with Adam (Kingma & Ba, 2015) with initial learning rate $10^{-3}$, batch size 128, and gradient clipping. We use macro-average score over all tasks as our validation metric, and perform a validation check every 10k updates. We divide the learning rate by 5 whenever validation performance does not improve. We stop training when the learning rate drops below $10^{-5}$ or performance does not improve after 5 validation checks.

### B.3   Sentence Representation Models

We evaluate the following sentence representation models:

1. CBoW, the average of the GloVe embeddings of the tokens in the sentence.
2. Skip-Thought (Kiros et al., 2015), a sequence-to-sequence(s) model trained to generate the previous and next sentences given the middle sentence. We use the original pre-trained model[7] trained on sequences of sentences from the Toronto Book Corpus (Zhu et al. 2015, TBC).
3. InferSent (Conneau et al., 2017), a BiLSTM with max-pooling trained on MNLI and SNLI.
4. DisSent (Nie et al., 2017), a BiLSTM with max-pooling trained to predict the discourse marker (*because*, *so*, etc.) relating two sentences on data derived from TBC. We use the variant trained for eight-way classification.
5. GenSen (Subramanian et al., 2018), a sequence-to-sequence model trained on a variety of supervised and unsupervised objectives. We use the variant of the model trained on both MNLI and SNLI, the Skip-Thought objective on TBC, and a constituency parsing objective on the Billion Word Benchmark.

We train task-specific classifiers on top of frozen sentence encoders, using the default parameters from SentEval. See https://github.com/nyu-mll/SentEval for details and code.

## C   Development Set Results

The GLUE website limits users to two submissions per day in order to avoid overfitting to the private test data. To provide a reference for future work on GLUE, we present the best development set results achieved by our baselines in Table 6.

## D   Benchmark Website Details

GLUE's online platform is built using React, Redux and TypeScript. We use Google Firebase for data storage and Google Cloud Functions to host and run our grading script when a submission is made. Figure 1 shows the visual presentation of our baselines on the leaderboard.

---

[7]`github.com/ryankiros/skip-thoughts`

| Rank | Name | Model | URL | Score | CoLA | SST-2 | MRPC | STS-B | QQP | MNLI-m | MNLI-mm | QNLI | RTE | WNLI |
|---|---|---|---|---|---|---|---|---|---|---|---|---|---|---|
| 1 | GLUE Baselines | BiLSTM+ELMo+Attn | | 68.9 | 18.9 | 91.6 | 77.3/83.5 | 72.8/71.1 | 83.5/63.3 | 75.6 | 75.9 | 81.7 | 61.2 | 65.1 |
| | | GenSen | | 66.6 | 7.7 | 83.1 | 76.6/83.0 | 79.3/79.2 | 82.9/59.8 | 71.4 | 71.3 | 82.3 | 59.2 | 65.1 |
| | | Single Task BiLSTM+ELMo | | 66.2 | 35.0 | 90.2 | 69.0/80.8 | 64.0/60.2 | 85.7/65.6 | 72.9 | 73.4 | 69.4 | 50.1 | 65.1 |
| | | BiLSTM+Attn | | 65.7 | 0.0 | 85.0 | 75.1/83.7 | 73.9/71.8 | 84.3/63.6 | 72.2 | 72.1 | 82.1 | 61.7 | 63.7 |
| | | BiLSTM+ELMo | | 64.9 | 27.5 | 89.6 | 76.2/83.5 | 67.0/65.9 | 78.5/57.8 | 67.1 | 68.0 | 66.7 | 55.7 | 62.3 |
| | | Single Task BiLSTM+ELMo+Attn | | 64.8 | 35.0 | 90.2 | 68.8/80.2 | 55.5/52.5 | 86.5/66.1 | 76.9 | 76.7 | 61.1 | 50.3 | 65.1 |
| | | InferSent | | 64.7 | 4.5 | 85.1 | 74.1/81.2 | 75.9/75.3 | 81.7/59.1 | 66.1 | 65.7 | 79.8 | 58.0 | 65.1 |
| | | BiLSTM+CoVe+Attn | | 64.3 | 19.4 | 83.6 | 75.2/83.0 | 72.3/71.1 | 84.9/61.1 | 69.9 | 68.7 | 78.9 | 38.3 | 65.1 |
| | | BiLSTM | | 63.5 | 24.0 | 85.8 | 71.9/82.1 | 68.8/67.0 | 80.2/59.1 | 65.8 | 66.0 | 71.1 | 46.8 | 63.7 |
| | | Single Task BiLSTM+CoVe | | 62.4 | 14.5 | 88.5 | 73.4/81.4 | 67.2/64.1 | 83.3/59.4 | 64.5 | 64.8 | 64.8 | 53.5 | 61.6 |
| | | BiLSTM+CoVe | | 62.2 | 16.2 | 84.3 | 71.8/80.0 | 68.0/67.1 | 82.0/59.1 | 65.3 | 65.9 | 70.4 | 44.2 | 65.1 |
| | | DisSent | | 62.1 | 4.9 | 83.7 | 74.1/81.7 | 66.1/64.8 | 82.6/59.5 | 58.7 | 59.1 | 75.2 | 56.4 | 65.1 |
| | | Single Task BiLSTM | | 62.0 | 15.7 | 85.9 | 69.3/79.4 | 66.0/62.8 | 81.7/61.4 | 70.3 | 70.8 | 60.8 | 52.8 | 62.3 |
| | | Skip-Thought | | 61.5 | 0.0 | 81.8 | 71.7/80.8 | 71.8/69.7 | 82.2/56.4 | 62.9 | 62.8 | 74.7 | 53.1 | 65.1 |
| | | Single Task BiLSTM+CoVe+Attn | | 60.8 | 14.5 | 88.5 | 68.6/79.7 | 57.2/53.6 | 84.1/60.1 | 71.6 | 71.5 | 53.8 | 52.7 | 64.4 |
| | | Single Task BiLSTM+Attn | | 60.0 | 15.7 | 85.9 | 68.5/80.3 | 59.3/55.8 | 83.5/62.9 | 74.2 | 73.8 | 51.9 | 51.9 | 55.5 |
| | | CBOW | | 58.9 | 0.0 | 80.0 | 73.4/81.5 | 61.2/58.7 | 79.1/51.4 | 56.0 | 56.4 | 75.1 | 54.1 | 62.3 |

Figure 1: The benchmark website leaderboard. An expanded view shows additional details about each submission, including a brief prose description and parameter count.

| Category | Count | % Neutral | % Contradiction | % Entailment |
|---|---|---|---|---|
| Lexical Semantics | 368 | 31.0 | 27.2 | 41.8 |
| Predicate-Argument Structure | 424 | 37.0 | 13.7 | 49.3 |
| Logic | 364 | 37.6 | 26.9 | 35.4 |
| Knowledge | 284 | 26.4 | 31.7 | 41.9 |

Table 7: Diagnostic dataset statistics by coarse-grained category. Note that some examples may be tagged with phenomena belonging to multiple categories.

# E  ADDITIONAL DIAGNOSTIC DATA DETAILS

The dataset is designed to allow for analyzing many levels of natural language understanding, from word meaning and sentence structure to high-level reasoning and application of world knowledge. To make this kind of analysis feasible, we first identify four broad categories of phenomena: Lexical Semantics, Predicate-Argument Structure, Logic, and Knowledge. However, since these categories are vague, we divide each into a larger set of fine-grained subcategories. Descriptions of all of the fine-grained categories are given in the remainder of this section. These categories are just one lens that can be used to understand linguistic phenomena and entailment, and there is certainly room to argue about how examples should be categorized, what the categories should be, etc. These categories are not based on any particular linguistic theory, but broadly based on issues that linguists have often identified and modeled in the study of syntax and semantics.

The dataset is provided not as a benchmark, but as an analysis tool to paint in broad strokes the kinds of phenomena a model may or may not capture, and to provide a set of examples that can serve for error analysis, qualitative model comparison, and development of adversarial examples that expose a model's weaknesses. Because the distribution of language is somewhat arbitrary, it will not be helpful to compare performance of the same model on different categories. Rather, we recommend comparing performance that different models score on the same category, or using the reported scores as a guide for error analysis.

We show coarse-grain category counts and label distributions of the diagnostic set in Table 7.

## E.1 LEXICAL SEMANTICS

These phenomena center on aspects of word meaning.

**Lexical Entailment**  Entailment can be applied not only on the sentence level, but the word level. For example, we say "dog" lexically entails "animal" because anything that is a dog is also an animal, and "dog" lexically contradicts "cat" because it is impossible to be both at once. This relationship applies to many types of words (nouns, adjectives, verbs, many prepositions, etc.) and the relationship between lexical and sentential entailment has been deeply explored, e.g., in systems of natural logic. This connection often hinges on monotonicity in language, so many Lexical Entailment examples will also be tagged with one of the Monotone categories, though we do not do this in every case (see Monotonicity, under Logic).

**Morphological Negation**  This is a special case of lexical contradiction where one word is derived from the other: from "affordable" to "unaffordable", "agree" to "disagree", etc. We also include examples like "ever" and "never". We also label these examples with Negation or Double Negation, since they can be viewed as involving a word-level logical negation.

**Factivity**  Propositions appearing in a sentence may be in any entailment relation with the sentence as a whole, depending on the context in which they appear. In many cases, this is determined by lexical triggers (usually verbs or adverbs) in the sentence. For example,

- "I recognize that X" entails "X".
- "I did not recognize that X" entails "X".
- "I believe that X" does not entail "X".
- "I am refusing to do X" contradicts "I am doing X".
- "I am not refusing to do X" does not contradict "I am doing X".
- "I almost finished X" contradicts "I finished X".
- "I barely finished X" entails "I finished X".

Constructions like "I recognize that X" are often called factive, since the entailment (of X above, regarded as a presupposition) persists even under negation. Constructions like "I am refusing to do X" above are often called implicative, and are sensitive to negation. There are also cases where a sentence (non-)entails the existence of an entity mentioned in it, for example "I have found a unicorn" entails "A unicorn exists" while "I am looking for a unicorn" doesn't necessarily entail "A unicorn exists". Readings where the entity does not necessarily exist are often called intensional readings, since they seem to deal with the properties denoted by a description (its intension) rather than being reducible to the set of entities that match the description (its extension, which in cases of non-existence will be empty).

We place all examples involving these phenomena under the label of Factivity. While it often depends on context to determine whether a nested proposition or existence of an entity is entailed by the overall statement, very often it relies heavily on lexical triggers, so we place the category under Lexical Semantics.

**Symmetry/Collectivity**  Some propositions denote symmetric relations, while others do not. For example, "John married Gary" entails "Gary married John" but "John likes Gary" does not entail "Gary likes John". Symmetric relations can often be rephrased by collecting both arguments into the subject: "John met Gary" entails "John and Gary met". Whether a relation is symmetric, or admits collecting its arguments into the subject, is often determined by its head word (e.g., "like", "marry" or "meet"), so we classify it under Lexical Semantics.

**Redundancy**  If a word can be removed from a sentence without changing its meaning, that means the word's meaning was more-or-less adequately expressed by the sentence; so, identifying these cases reflects an understanding of both lexical and sentential semantics.

**Named Entities**    Words often name entities that exist in the world. There are many different kinds of understanding we might wish to understand about these names, including their compositional structure (for example, the "Baltimore Police" is the same as the "Police of the City of Baltimore") or their real-world referents and acronym expansions (for example, "SNL" is "Saturday Night Live"). This category is closely related to World Knowledge, but focuses on the semantics of names as lexical items rather than background knowledge about their denoted entities.

**Quantifiers**    Logical quantification in natural language is often expressed through lexical triggers such as "every", "most", "some", and "no". While we reserve the categories in Quantification and Monotonicity for entailments involving operations on these quantifiers and their arguments, we choose to regard the interchangeability of quantifiers (e.g., in many cases "most" entails "many") as a question of lexical semantics.

### E.2    PREDICATE-ARGUMENT STRUCTURE

An important component of understanding the meaning of a sentence is understanding how its parts are composed together into a whole. In this category, we address issues across that spectrum, from syntactic ambiguity to semantic roles and coreference.

**Syntactic Ambiguity: Relative Clauses, Coordination Scope**    These two categories deal purely with resolving syntactic ambiguity. Relative clauses and coordination scope are both sources of a great amount of ambiguity in English.

**Prepositional phrases**    Prepositional phrase attachment is a particularly difficult problem that syntactic parsers in NLP systems continue to struggle with. We view it as a problem both of syntax and semantics, since prepositional phrases can express a wide variety of semantic roles and often semantically apply beyond their direct syntactic attachment.

**Core Arguments**    Verbs select for particular arguments, particularly subjects and objects, which might be interchangeable depending on the context or the surface form. One example is the ergative alternation: "Jake broke the vase" entails "the vase broke" but "Jake broke the vase" does not entail "Jake broke". Other rearrangements of core arguments, such as those seen in Symmetry/Collectivity, also fall under the Core Arguments label.

**Alternations: Active/Passive, Genitives/Partitives, Nominalization, Datives**    All four of these categories correspond to syntactic alternations that are known to follow specific patterns in English:

- Active/Passive: "I saw him" is equivalent to "He was seen by me" and entails "He was seen".

- Genitives/Partitives: "the elephant's foot" is the same thing as "the foot of the elephant".

- Nominalization: "I caused him to submit his resignation" entails "I caused the submission of his resignation".

- Datives: "I baked him a cake" entails "I baked a cake for him" and "I baked a cake" but not "I baked him".

**Ellipsis/Implicits**    Often, the argument of a verb or other predicate is omitted (elided) in the text, with the reader filling in the gap. We can construct entailment examples by explicitly filling in the gap with the correct or incorrect referents. For example, the premise "Putin is so entrenched within Russias ruling system that many of its members can imagine no other leader" entails "Putin is so entrenched within Russias ruling system that many of its members can imagine no other leader than Putin" and contradicts "Putin is so entrenched within Russias ruling system that many of its members can imagine no other leader than themselves."

This is often regarded as a special case of anaphora, but we decided to split out these cases from explicit anaphora, which is often also regarded as a case of coreference (and attempted to some degree in modern coreference resolution systems).

**Anaphora/Coreference**   Coreference refers to when multiple expressions refer to the same entity or event. It is closely related to Anaphora, where the meaning of an expression depends on another (antecedent) expression in context. These two phenomena have significant overlap; for example, pronouns ("she", "we", "it") are anaphors that are co-referent with their antecedents. However, they also may occur independently, such as coreference between two definite noun phrases (e.g., "Theresa May "and the "British Prime Minister") that refer to the same entity, or anaphora from a word like "other" which requires an antecedent to distinguish something from. In this category we only include cases where there is an explicit phrase (anaphoric or not) that is co-referent with an antecedent or other phrase. We construct examples for these in much the same way as for Ellipsis/Implicits.

**Intersectivity**   Many modifiers, especially adjectives, allow non-intersective uses, which affect their entailment behavior. For example:

- Intersective: "He is a violinist and an old surgeon" entails "He is an old violinist" and "He is a surgeon".

- Non-intersective: "He is a violinist and a skilled surgeon" does not entail "He is a skilled violinist".

- Non-intersective: "He is a fake surgeon" does not entail "He is a surgeon".

Generally, an intersective use of a modifier, like "old" in "old men", is one which may be interpreted as referring to the set of entities with both properties (they are old and they are men). Linguists often formalize this using set intersection, hence the name.

Intersectivity is related to Factivity. For example, "fake" may be regarded as a counter-implicative modifier, and these examples will be labeled as such. However, we choose to categorize intersectivity under predicate-argument structure rather than lexical semantics, because generally the same word will admit both intersective and non-intersective uses, so it may be regarded as an ambiguity of argument structure.

**Restrictivity**   Restrictivity is most often used to refer to a property of uses of noun modifiers. In particular, a restrictive use of a modifier is one that serves to identify the entity or entities being described, whereas a non-restrictive use adds extra details to the identified entity. The distinction can often be highlighted by entailments:

- Restrictive: "I finished all of my homework due today" does not entail "I finished all of my homework".

- Non-restrictive: "I got rid of all those pesky bedbugs" entails "I got rid of all those bedbugs".

Modifiers that are commonly used non-restrictively are appositives, relative clauses starting with "which" or "who", and expletives (e.g. "pesky"). Non-restrictive uses can appear in many forms.

E.3   LOGIC

With an understanding of the structure of a sentence, there is often a baseline set of shallow conclusions that can be drawn using logical operators and often modeled using the mathematical tools of logic. Indeed, the development of mathematical logic was initially guided by questions about natural language meaning, from Aristotelian syllogisms to Fregean symbols. The notion of entailment is also borrowed from mathematical logic.

**Propositional Structure: Negation, Double Negation, Conjunction, Disjunction, Conditionals**
All of the basic operations of propositional logic appear in natural language, and we tag them where they are relevant to our examples:

- Negation: "The cat sat on the mat" contradicts "The cat did not sit on the mat".

- Double negation: "The market is not impossible to navigate" entails "The market is possible to navigate".

- Conjunction: "Temperature and snow consistency must be just right" entails "Temperature must be just right".
- Disjunction: "Life is either a daring adventure or nothing at all" does not entail, but is entailed by, "Life is a daring adventure".
- Conditionals: "If both apply, they are essentially impossible" does not entail "They are essentially impossible".

Conditionals are more complicated because their use in language does not always mirror their meaning in logic. For example, they may be used at a higher level than the at-issue assertion: "If you think about it, it's the perfect reverse psychology tactic" entails "It's the perfect reverse psychology tactic".

**Quantification: Universal, Existential** Quantifiers are often triggered by words such as "all", "some", "many", and "no". There is a rich body of work modeling their meaning in mathematical logic with generalized quantifiers. In these two categories, we focus on straightforward inferences from the natural language analogs of universal and existential quantification:

- Universal: "All parakeets have two wings" entails, but is not entailed by, "My parakeet has two wings".
- Existential: "Some parakeets have two wings" does not entail, but is entailed by, "My parakeet has two wings".

**Monotonicity: Upward Monotone, Downward Monotone, Non-Monotone** Monotonicity is a property of argument positions in certain logical systems. In general, it gives a way of deriving entailment relations between expressions that differ on only one subexpression. In language, it can explain how some entailments propagate through logical operators and quantifiers.

For example, "pet" entails "pet squirrel", which further entails "happy pet squirrel". We can demonstrate how the quantifiers "a", "no" and "exactly one" differ with respect to monotonicity:

- "I have a pet squirrel" entails "I have a pet", but not "I have a happy pet squirrel".
- "I have no pet squirrels" does not entail "I have no pets", but does entail "I have no happy pet squirrels".
- "I have exactly one pet squirrel" entails neither "I have exactly one pet" nor "I have exactly one happy pet squirrel".

In all of these examples, "pet squirrel" appears in what we call the restrictor position of the quantifier. We say:

- "a" is upward monotone in its restrictor: an entailment in the restrictor yields an entailment of the whole statement.
- "no" is downward monotone in its restrictor: an entailment in the restrictor yields an entailment of the whole statement in the opposite direction.
- "exactly one" is non-monotone in its restrictor: entailments in the restrictor do not yield entailments of the whole statement.

In this way, entailments between sentences that are built off of entailments of sub-phrases almost always rely on monotonicity judgments; see, for example, Lexical Entailment. However, because this is such a general class of sentence pairs, to keep the Logic category meaningful we do not always tag these examples with monotonicity.

**Richer Logical Structure: Intervals/Numbers, Temporal** Some higher-level facets of reasoning have been traditionally modeled using logic, such as actual mathematical reasoning (entailments based off of numbers) and temporal reasoning (which is often modeled as reasoning about a mathematical timeline).

- Intervals/Numbers: "I have had more than 2 drinks tonight" entails "I have had more than 1 drink tonight".
- Temporal: "Mary left before John entered" entails "John entered after Mary left".

### E.4   KNOWLEDGE

Strictly speaking, world knowledge and common sense are required on every level of language understanding for disambiguating word senses, syntactic structures, anaphora, and more. So our entire suite (and any test of entailment) does test these features to some degree. However, in these categories, we gather examples where the entailment rests not only on correct disambiguation of the sentences, but also application of extra knowledge, whether concrete knowledge about world affairs or more common-sense knowledge about word meanings or social or physical dynamics.

**World Knowledge**   In this category we focus on knowledge that can clearly be expressed as facts, as well as broader and less common geographical, legal, political, technical, or cultural knowledge. Examples:

- "This is the most oniony article I've seen on the entire internet" entails "This article reads like satire".
- "The reaction was strongly exothermic" entails "The reaction media got very hot".
- "There are amazing hikes around Mt. Fuji" entails "There are amazing hikes in Japan" but not "There are amazing hikes in Nepal".

**Common Sense**   In this category we focus on knowledge that is more difficult to express as facts and that we expect to be possessed by most people independent of cultural or educational background. This includes a basic understanding of physical and social dynamics as well as lexical meaning (beyond simple lexical entailment or logical relations). Examples:

- "The announcement of Tillerson's departure sent shock waves across the globe" contradicts "People across the globe were prepared for Tillerson's departure".
- "Marc Sims has been seeing his barber once a week, for several years" entails "Marc Sims has been getting his hair cut once a week, for several years".
- "Hummingbirds are really attracted to bright orange and red (hence why the feeders are usually these colours)" entails "The feeders are usually coloured so as to attract hummingbirds".

