# OpenReview forum: "GLUE: A Multi-Task Benchmark and Analysis Platform for Natural Language Understanding"
_ICLR.cc/2019/Conference_

### Official Review · AnonReviewer1 · 2018-11-01
**A timely and useful resource**

**Rating:** 8
**Confidence:** 4

**Review:**

This paper introduces the General Language Understanding Evaluation (GLUE) benchmark and platform, which aims to evaluate representations of language with an emphasis on generalizability. This is a timely contribution and GLUE will be an impactful resource for the NLP community. This is mitigated, perhaps, somewhat by the recent release of decaNLP. But, as discussed the authors, this has a different focus (re-framing all tasks as QQ) and further does not feature the practical tools released here (leaderboard, error analysis) that will help drive progress.

Some comments below.

- The inclusion of the small diagnostic dataset was a nice addition and it would be nice if future corpora included similar.

- Implicit in this and related efforts is the assumption that parameter sharing ought to be possible and fruitful across even quite diverse tasks. While I do not object to this, it would be nice if the authors could make an explicit case here as to why should we believe this to be the case.

- The proposed platform is touted as one of the main contributions here, but not pointed to -- I assume for anonymity preserving reasons, but still would have been nice for this to be made explicit.

- I would consider pushing Table 5 (Appendix) into the main text.

---

> ### Author Response · Authors · 2018-11-19
> **Author response**
>
> Thank you for your review!
>
> We agree that the diagnostic data is a key contribution of our work. We wanted to not only have an application-driven measure of progress, but also a targeted measure of performance on specific natural language phenomena that we would expect a general-purpose NLU model to handle well.
>
> Regarding parameter sharing, our intent was to include tasks with very little training data such that automated systems could not do well learning on just those tasks’ data. Competitive systems, then, would need to include some form of knowledge-sharing from outside data.  In only requiring model predictions to evaluate on test, we wanted to avoid restricting future research to any particular paradigm of knowledge sharing. We use multi-task learning and parameter sharing because it is a straightforward baseline with lots of precedent (GenSen, Collobert and Weston, etc.), so we thought it necessary to include.
>
> Could you please clarify how a small *test* set would encourage few-shot learning? To the best of our knowledge, few-shot learning is when you have a small *training* set for the target task.
>
> Re: table 5, we agree! We’ll post an updated version of the paper shortly.

---

### Official Review · AnonReviewer2 · 2018-11-02
**Weak reject**

**Rating:** 5
**Confidence:** 2

**Review:**

The paper proposes a new benchmark for natural language understanding: GLUE. Models will be evaluated based on a diverse set of existing language understanding tasks which encourages the models to learn shared knowledge across different tasks. The authors empirically show that models trained with multiple tasks in the dataset perform better than models that focused on one specific task. They also point out existing methods are not able achieve good performance in this dataset and request for more general natural language understanding system. The work also collects an expert evaluated diagnostic evaluation dataset for further examination for the models.

Quality: borderline, clarity:good, originality: borderline, significance: good,

Pros:
- The benchmark is set up in a online platform with leaderboard which can be easily accessible to people.
- The benchmark comes with a diagnostic evaluation dataset with coarse-grained and fine-grained categories that examine different aspect of language understanding abilities.
- Baseline results for major existing models are provided

Cons:
- The author should provide more detailed analysis and interpretable explanations for the results as opposed to simply stating that the overall performance is better.
For example, why attention hurts performance in single task training? Why multi-tasks training actually leads to worse performance on some of the dataset? Do these phenomenons still exist if you train on a different subset of the dataset?
What are the samples that the models failed to perform well? It would be nice to get some more insights and conclusions based on the results obtained from this benchmark to shed some lights on how to improve these models. The results section should be seriously revised.

- The diagnostic evaluation dataset seems to be a way to better understand the model, however, it is hard to see the scope of the data (are the samples under each categories balanced?). Besides, the examples in the dataset seems very confusing even for humans (Table 2).  The evaluation with NLP expert is also far from perfect. I wonder how accurate is this dataset annotated (or even the sentences make sense or not), and how suitable it is for evaluating model’s language understanding abilities. It would be nice if the authors can include some statistics about the dataset.

The paper proposes a useful benchmark that measures different aspects of language understanding abilities which would be helpful to the community. However, I feel the novelty or take away messages from the experiment section is limited.

---

> ### Author Response · Authors · 2018-11-19
> **Author response**
>
> Thanks for your thoughtful review!
>
> While we agree that more analysis would be nice, the central contribution of our paper is to motivate and introduce the benchmark. Thus, our experiments are designed to give baseline numbers for a broad range of models and to highlight the benefits of our design decisions, in order to make the case that our benchmark offers useful improvements over previous evaluation standards like SentEval.
>
> Regarding the diagnostic data, we do believe much of the information you mention is present in the paper. For example, we explicitly give the class distribution for the entire dataset (including statistics by category is a good point - we will soon add the class distribution per coarse-grained category), expert annotator agreement (high, at \kappa = 0.73), and human performance (R_3 = 0.8 versus 0.28 for the best model). The last statistic in particular indicates that these examples are understandable and solvable by humans and challenging for existing models. These numbers are in-line with other semantic datasets that have been productively used by the community, for example SQuADv2 (humans get ~87 EM); SimLex-999 (0.67 correlation); WordSimilarity-353 (.61 correlation). We agree that human annotations are not perfect, but perfect annotations don’t exist, and datasets can still be useful even when their human annotations are a little noisy, as in the previously mentioned examples.

---

### Official Review · AnonReviewer3 · 2018-11-05
**Interesting new benchmark**

**Rating:** 7
**Confidence:** 1

**Review:**

Summary:
GLUE is a benchmark consisting of multiple natural language understanding tasks
that functions via uploading to a website and receiving a score based on
privately held test set labels.
Tasks include acceptability judgement, sentiment prediction, semantic equivalence
detection, judgement of premise hypothesis entailment, question paragraph pair
matching, etc..
The benchmark also includes a diagnostic dataset with logical tasks such as
lexical entailment and understanding quantifiers.

In addition to presenting the benchmark itself, the paper also presents models
for performance baselines.
There is some brief analysis of the ability of Sentence2Vector vs. more complex
models with e.g. attention mechanisms and of single-task vs. multi-task training.

Evaluation:
The GLUE benchmark seems like a well designed benchmark that could potentially
ignite new progress in the area of NLU.
But since I'm not an expert in the area of language modeling and know almost
nothing about existing benchmarks I cannot validate the added benefit over
existing benchmarks and the novelty of the suggested benchmarking approach.

Details:
The paper is well written, clear and easy to follow.

The proposed benchmarks seem reasonable and illustrate the difficulty of
benchmark tasks that involve logical structure.

Page 5: showing showing (Typo)

---

> ### Author Response · Authors · 2018-11-19
> **Author response**
>
> Thank you for your review!
>
> To clarify, GLUE is not a benchmark for language modeling (the task of modeling the probability of a piece of text) but rather (classification or regression based) natural language understanding.
>
> Regarding previous work in this space, we mention in the paper what we believe are the two major comparable works: SentEval and DecaNLP, and highlight the benefits over these neighbors afforded by our design decisions.  As your review hints at, we believe that a major benefit of a well-designed benchmark is that it can more clearly distinguish models that make significant improvements and thereby incentivize researchers to work on it. Early results on GLUE seem to suggest that we have been successful in that regard.

---

### Public Comment · ~quan_vuong1 · 2018-10-08
**Clarification Questions**

Thank you for the paper and the dataset!

I'm using GLUE in my own research and would like to ask a few clarification questions.

In Table 1, why is it that only accuracy (and not F1) is used to measure performance on SST-2, QNLI, RTE and WNLI?

For QNLI, in the sentence, "The task is to determine whether the context sentence contains the answer to the question.", is the task is to then:

- Determine whether the answer is contained in any context sentence. The label for a (context, question) pair would then be binary, where 1 indicates at least one sentence in the context contains the answer to the question and 0 otherwise.
OR
- Determine the sentence that contains the answer out of all the sentences in the context passage. The label for each sentence in the context passage would then be binary, i.e. a sentence is assigned a gold label of 0 if the answer is not part of the sentence and 1 if it is.

---

> ### Author Response · Authors · 2018-10-10
> **Clarification Answers**
>
> Hi Quan,
> Thanks for using GLUE! Regarding your questions:
>
> - SST-2, QNLI, RTE, and WNLI are all roughly balanced, so using accuracy is reasonable here (and has been used as the evaluation metric in the past). We use F1 for datasets with class imbalances, and also to maintain comparability with previous work on those datasets.
>
> - For QNLI, each example consists of the original question and a single sentence from the context paragraph that was originally paired with that question. The task is to determine if that context sentence contains an answering span to the question. If I understand correctly, it's the latter of the two options you mentioned, but we do some additional filtering of question, context sentence pairs that are too easy (due to low lexical overlap).
>
> Let us know if you have any additional questions.

---

### Meta-Review · Area_Chair1 · 2018-12-18
**Multitask learning is one of the most important problems in AI**

**Confidence:** 5
**Recommendation:** Accept (Poster)

**Metareview:**

This paper provides an interesting benchmark for multitask learning in NLP.
I wish the dataset included language generation tasks instead of just classification but it's still a step in the right direction.